# A test of the pioneer factor hypothesis using ectopic liver gene activation

Jeffrey L Hansen[1,2,3], Kaiser J Loell[1,2], Barak A Cohen[1,2]*

[1]Edison Center for Genome Sciences and Systems Biology, Washington University in St. Louis, St. Louis, United States; [2]Department of Genetics, Washington University in St. Louis, St. Louis, United States; [3]Medical Scientist Training Program, Washington University in St. Louis, St. Louis, United States

**Abstract** The pioneer factor hypothesis (PFH) states that pioneer factors (PFs) are a subclass of transcription factors (TFs) that bind to and open inaccessible sites and then recruit non-pioneer factors (non-PFs) that activate batteries of silent genes. The PFH predicts that ectopic gene activation requires the sequential activity of qualitatively different TFs. We tested the PFH by expressing the endodermal PF FOXA1 and non-PF HNF4A in K562 lymphoblast cells. While co-expression of FOXA1 and HNF4A activated a burst of endoderm-specific gene expression, we found no evidence for a functional distinction between these two TFs. When expressed independently, both TFs bound and opened inaccessible sites, activated endodermal genes, and 'pioneered' for each other, although FOXA1 required fewer copies of its motif for binding. A subset of targets required both TFs, but the predominant mode of action at these targets did not conform to the sequential activity predicted by the PFH. From these results, we hypothesize an alternative to the PFH where 'pioneer activity' depends not on categorically different TFs but rather on the affinity of interaction between TF and DNA.

## Editor's evaluation

The article elegantly tests a leading hypothesis in the field and demonstrates that the pioneer factor model is not sufficient to explain most of the gene activation when liver transcription factors (TFs) are ectopically expressed. These results provide a new standard for testing the degree to which a TF acts as a pioneer factor. Moreover, they suggest that cis elements (i.e., motif density) are critical to open a locus upon transcription factor expression. How motif density and TF binding contribute to nucleosome eviction will be interesting to unravel in future studies.

## Introduction

Transcription factors (TFs) face steric hindrance when instances of their motifs are occluded by nucleosomes (*Kornberg, 1974*; *Kaplan et al., 2009*). This barrier prevents spurious transcription but must be overcome during development when TFs activate batteries of silent genes. The pioneer factor hypothesis (PFH) describes how TFs recognize and activate nucleosome-occluded targets. According to the PFH, categorically different TFs cooperate sequentially to activate their targets. Pioneer factors (PFs) bind to and open inaccessible sites and then recruit non-pioneer factors (non-PFs) that are responsible for recruiting additional factors to initiate gene expression (*McPherson et al., 1993*; *Shim et al., 1998*; *Cirillo et al., 1998*; *Cirillo et al., 2002*).

PFs also play a primary role in cellular reprogramming by first engaging silent regulatory sites of ectopic lineages (*Iwafuchi-Doi and Zaret, 2014*). Continuous overexpression of PFs and non-PFs can lead to a variety of lineage conversions (*Wapinski et al., 2013*; *Matsuda et al., 2019*; *Soufi*

*For correspondence: cohen@genetics.wustl.edu

Competing interest: The authors declare that no competing interests exist.

**eLife digest** Cells only use a fraction of their genetic information to make the proteins they need. The rest is carefully packaged away and tightly bundled in structures called nucleosomes. This physically shields the DNA from being accessed by transcription factors – the molecular actors that can read genes and kickstart the protein production process. Effectively, the genetic sequences inside nucleosomes are being silenced.

However, during development, transcription factors must overcome this nucleosome barrier and activate silent genes to program cells. The pioneer factor hypothesis describes how this may be possible: first, 'pioneer' transcription factors can bind to and 'open up' nucleosomes to make target genes accessible. Then, non-pioneer factors can access the genetic sequence and recruit cofactors that begin copying the now-exposed genetic information.

The widely accepted theory is based on studies of two proteins – FOXA1, an archetypal pioneer factor, and HNF4A, a non-pioneer factor – but the predictions of the pioneer factor hypothesis have yet to be explicitly tested. To do so, Hansen et al. expressed FOXA1 and HNF4A, separately and together, in cells which do not usually make these proteins. They then assessed how the proteins could bind to DNA and impact gene accessibility and transcription.

The experiments demonstrate that FOXA1 and HNF4A do not necessarily follow the two-step activation predicted by the pioneer factor hypothesis. When expressed independently, both transcription factors bound and opened inaccessible sites, activated target genes, and 'pioneered' for each other. Similar patterns were observed across the genome. The only notable distinction between the two factors was that FOXA1, the archetypal pioneering factor, required fewer copies of its target sequence to bind DNA than HNF4A.

These findings led Hansen et al. to propose an alternative theory to the pioneer factor hypothesis which eliminates the categorical distinction between pioneer and non-pioneer factors. Overall, this work has implications for how biologists understand the way that transcription factors activate silent genes during development.

*et al., 2015*; *Soufi et al., 2012*; *Sekiya and Suzuki, 2011*; *Morris et al., 2014*). The conversion from embryonic fibroblasts to induced endoderm progenitors offers one clear example (*Sekiya and Suzuki, 2011*; *Morris et al., 2014*). This reprogramming cocktail combines the canonical PF FOXA1 (*Cirillo et al., 2002*) and non-PF HNF4A (*Karagianni et al., 2020*) and is suggested to rely upon the sequential activity of FOXA1 followed by HNF4A (*Horisawa et al., 2020*).

The PFH makes strong predictions about the activities of ectopically expressed PFs and non-PFs. Because PFs are defined by their ability to bind nucleosome-occluded instances of their motifs, the PFH predicts that PFs should bind to a large fraction of their motifs. However, similar to other TFs, PFs only bind a limited subset of their inaccessible motifs (*Barozzi et al., 2014*; *Mayran et al., 2018*; *Donaghey et al., 2018*; *Manandhar et al., 2017*). There are chromatin states that are prohibitive to PF binding (*Mayran et al., 2018*; *Zaret and Mango, 2016*), and, in at least two cases, FOXA1 requires help from other TFs to bind at its sites (*Donaghey et al., 2018*; *Swinstead et al., 2016*). These examples suggest that PFs are not always sufficient to open inaccessible chromatin. The PFH also predicts that non-PFs should only bind at accessible sites, yet the bacterial protein LexA can pioneer inaccessible sites in mammalian cells (*Miller and Widom, 2003*). These observations, and the absence of direct genome-wide interrogations of the PFH, prompted us to design experiments to test major predictions made by the PFH using FOXA1 and HNF4A as a model PF and non-PF.

To test these predictions, we expressed FOXA1 and HNF4A separately and together in K562 lymphoblast cells and then measured their effects on DNA-binding, chromatin accessibility, and gene activation. In contrast to the predictions of the PFH, we found that both FOXA1 and HNF4A could independently bind to inaccessible instances of their motifs, induce chromatin accessibility, and activate endoderm-specific gene expression. The only notable distinction between the two factors was that HNF4A required more copies of its motif to bind. When expressed together, co-binding could only be explained in a minority of cases by sequential FOXA1 and HNF4A activity. Instead, most co-bound sites required concurrent co-expression of both factors, which suggests cooperativity between these TFs at certain repressive genomic locations. We suggest that our findings present an alternative to

the PFH that eliminates the categorical distinction between PFs and non-PFs and instead posits that the energy required to pioneer occluded sites ('pioneer activity') comes from the affinity of interaction between TFs and DNA.

## Results

### Generation of FOXA1 and HNF4A clonal lines

We tested predictions of the PFH using FOXA1 as a model endoderm PF and HNF4A as a model non-PF. Because PFs are defined by their behavior in ectopic settings, we expressed FOXA1 and HNF4A in mesoderm-derived K562 lymphoblast cells. These cells express neither FOXA1 nor HNF4A and present a different complement of chromatin and cofactors. Thus, any ectopic signature that we observe is due primarily to the TFs themselves. We focused only on the initial response to TF expression to capture primary mechanisms of TF behavior and not the secondary effects that can lead to cellular conversion and that may confound our analyses.

To perform these experiments, we created lentiviruses that inducibly express either FOXA1 or HNF4A (*Figure 1A*). We created cassettes in which a doxycycline-inducible promoter drives either FOXA1 or HNF4A and cloned these cassettes separately into a lentiviral vector (*Meerbrey et al., 2011*) that constitutively expresses green fluorescent protein (GFP). Although PFs are typically expressed at supraphysiological levels (*Ng et al., 2021*; *Davis et al., 1987*), we infected K562 cells with each vector at a multiplicity of infection (MOI) of 1 to limit the degree of nonspecific effects. We then used flow cytometry to sort single cells and selected FOXA1 and HNF4A clones that had similar GFP levels to ensure that our clones carried a similar transgene load. Finally, we performed both doxycycline titration induction and time-course experiments to identify the minimum doxycycline concentration and treatment time for robust TF activity. We observed that 0.5 µg/ml doxycycline for 24 hr was the minimal treatment condition that allowed *FOXA1* and *HNF4A*, and their respective target genes *ALB* and *APOB*, to reach a plateau of expression (*Figure 1—figure supplement 1*). At this concentration, both *FOXA1* and *HNF4A* were induced approximately 1000-fold (*Figure 1—figure supplement 1*). We used these conditions in our subsequent experiments.

### Co-expression of FOXA1 and HNF4A in K562 cells conforms to the predictions of the PFH

The first prediction of the PFH is that co-expression of FOXA1 and HNF4A should be sufficient to induce ectopic tissue-specific gene expression. We tested this prediction by infecting our FOXA1 clonal line with HNF4A-expressing lentivirus to generate a double expression clonal line, hereafter referred to as FOXA1-HNF4A. Upon co-induction in K562 cells, we observed strong enrichment for both liver- and intestine-specific gene activation; FOXA1-HNF4A activated 91 liver-specific genes (18 expected, $p<10^{-38}$, cumulative hypergeometric) and 38 intestinal genes (9 expected by chance, $p<10^{-13}$, cumulative hypergeometric) (*Figure 1B*). The dual liver and intestine enrichment that we observed is consistent with the finding that intestinal gene regulatory networks appear during reprogramming experiments that aim to use FOXA1-HNF4A to convert embryonic fibroblasts to the liver lineage (*Morris et al., 2014*). We conclude that FOXA1 and HNF4A are sufficient to activate endoderm-specific gene expression in the ectopic K562 line.

Where ectopic genes are activated in K562 cells, the PFH predicts co-binding of FOXA1 and HNF4A at inaccessible sites and induction of chromatin accessibility. Alternatively, FOXA1 and HNF4A may not be able to overcome the K562 chromatin environment and instead activate gene expression by binding exclusively to accessible K562 sites. To distinguish between these possibilities, we measured FOXA1 and HNF4A binding by CUT&Tag (*Kaya-Okur et al., 2019*) after induction, and chromatin accessibility by ATAC-seq (*Buenrostro et al., 2015*) both before and after doxycycline induction. At the liver-specific locus *ALB*, FOXA1 and HNF4A co-bound at inaccessible sites and increased accessibility (*Figure 1C*). This pattern was consistent surrounding FOXA1-HNF4A-activated liver genes: 43 of the 53 co-bound sites within 50 kb of a FOXA1-HNF4A-activated gene were inaccessible prior to induction, and the accessibility signal at these co-bound sites increased substantially upon induction (*Figure 1D and E*).

Although we focused on functional binding surrounding activated liver genes, these patterns were consistent across the genome. The vast majority of both FOXA1 and HNF4A binding sites fell within

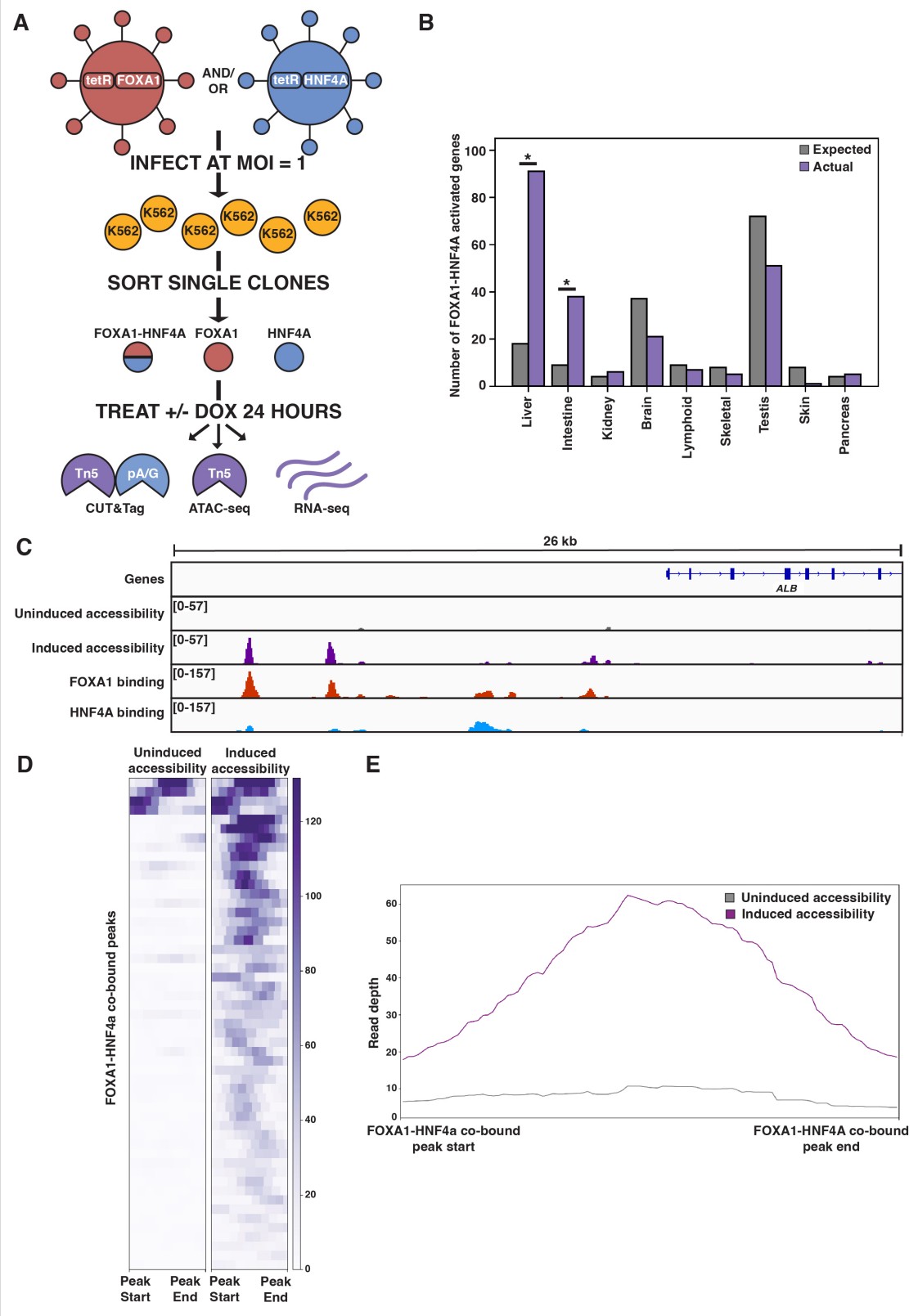

**Figure 1.** FOXA1-HNF4A pioneers liver-specific loci in K562 cells. (**A**) Schematic of experimental design to infect K562 cells with FOXA1- or HNF4A-lentivirus and then perform functional assays on dox-induced cells. In CUT&Tag, a protein A-protein G fusion (pA/G) increases the binding spectrum for Fc-binding and allows Tn5 recruitment to antibody-labeled transcription factor (TF) binding sites. In ATAC-seq, Tn5 homes to any accessible site. And in RNA-seq, polyA RNA is captured and sequenced. (**B**) The number of tissue-specific genes predicted from the hypergeometric distribution to

*Figure 1 continued*

be activated by FOXA1-HNF4A compared to the number actually activated. Both liver- (p<10⁻³⁸) and intestinal enrichment (p<10⁻¹³) are significant. There are 242 total liver-enriched genes and 122 total intestine-enriched genes. (**C**) Genome browser view of a representative liver-specific locus (*ALB*) in FOXA1-HNF4A clonal line that shows uninduced and induced accessibility, FOXA1 binding, and HNF4A binding. (**D**) Heatmap showing uninduced and induced accessibility at all FOXA1-HNF4A co-bound sites within 50 kb of each FOXA1-HNF4A-activated liver-specific gene (n = 53). (**E**) Meta plot showing average signal across each site from (**D**).

The online version of this article includes the following figure supplement(s) for figure 1:

**Figure supplement 1.** Titration of doxycycline concentration and treatment time for transcription factor (TF) and target gene induction.

**Figure supplement 2.** Characterization of FOXA1 and HNF4A binding patterns in FOXA1-HNF4A clone.

sites that were inaccessible prior to induction (-dox) (*Figure 1—figure supplement 2*), and both FOXA1 and HNF4A opened the majority of the inaccessible sites to which they bound (*Figure 1— figure supplement 2*). These results show that despite an entirely ectopic complement of chromatin and cofactors within mesoderm-derived K562 cells, the endodermal TFs FOXA1 and HNF4A can find and activate the correct genes. Most individual binding by FOXA1 and HNF4A near their co-activated genes occurred at the same sites bound in HepG2 liver cells (*Partridge et al., 2020*; *Figure 1—figure supplement 2*). Altogether we conclude that when co-expressed, FOXA1 and HNF4A conform to the predictions of the PFH and that cis-regulatory sequences are sufficient to guide their activity within an ectopic cell type.

## Both FOXA1 and HNF4A individually activate many liver-specific genes

We next sought to test whether ectopic tissue-specific gene expression in K562 cells results from the sequential activity of FOXA1 and HNF4A, as predicted by the PFH. The sequential activity model predicts that HNF4A will not bind to its sites without FOXA1, and that FOXA1 will not activate expression without HNF4A, such that neither FOXA1 nor HNF4A should activate tissue-specific gene expression when expressed alone. To test this prediction, we used the single-expression K562 lines to induce either FOXA1 or HNF4A alone and measured mRNA expression by RNA-seq. FOXA1 induction resulted in strong liver-specific enrichment (p<10⁻⁴, cumulative hypergeometric) and weak intestinal-specific enrichment (not significant) (*Figure 2A*), while HNF4A induction resulted in both strong liver-specific enrichment (p<10⁻⁸, cumulative hypergeometric) and strong intestinal-specific enrichment (p<10⁻¹⁵, cumulative hypergeometric) (*Figure 2B*). Importantly, neither FOXA1 nor HNF4A are expressed within K562 cells nor did they induce expression of the other TF, suggesting that the expression changes we observed were due to the independent effects of either FOXA1 or HNF4A.

When expressed individually, FOXA1 and HNF4A activated largely independent sets of liver genes (*Figure 2C*) and intestinal genes (*Figure 2D*). FOXA1 activates liver genes enriched for fibrinolysis and complement activation (*Supplementary file 1*), whereas HNF4A activates liver genes enriched for cholesterol import and lipoprotein remodeling (*Supplementary file 2*). Thus, in contrast to the predictions of the PFH, FOXA1 and HNF4A are each sufficient to induce separate and specific endodermal responses when expressed alone in K562 cells.

## Both FOXA1 and HNF4A can independently bind and open inaccessible sites around liver genes

Our results raised the possibility that both FOXA1 and HNF4A can bind and open inaccessible instances of their motifs. To test this, we induced FOXA1 and HNF4A expression individually and then measured each factor's binding profile and their accessibility profiles before and after induction. FOXA1 induction resulted in FOXA1 binding and induced accessibility adjacent to *ARG1*, a liver-specific gene that is silent in K562 cells (*Figure 3A*), while HNF4A alone bound and induced accessibility at sites nearby the liver-specific gene *APOC3* (*Figure 3B*). This pattern was consistent across liver-specific loci. 34 of the 59 FOXA1 binding sites within 50 kb of a FOXA1-activated liver gene were inaccessible and opened upon induction (*Figure 3C and E*) as was the case for 39 of the 76 HNF4A binding sites (*Figure 3D and F*). We observed similar patterns genome-wide. FOXA1 and HNF4A bound primarily to sites that were inaccessible prior to induction (-dox) (*Figure 3—figure supplement 1*), opened them (*Figure 3—figure supplement 1*), and in regions surrounding activated genes, most binding

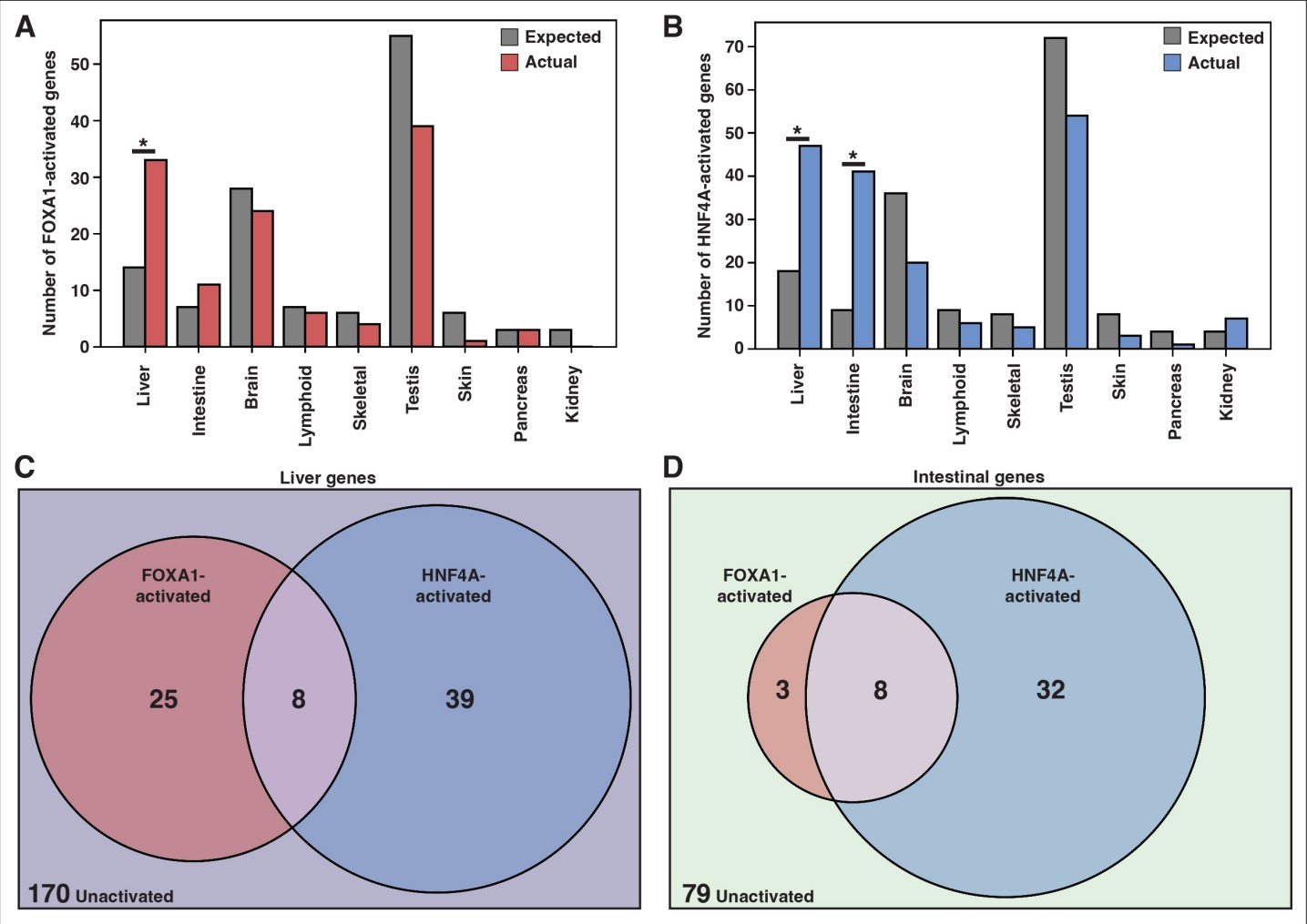

**Figure 2.** FOXA1 and HNF4A activate independent liver- and intestine-specific genes. (**A**) The number of tissue-specific genes predicted from the hypergeometric distribution to be activated by FOXA1 compared to the number actually activated. Liver enrichment (p<10⁻⁴) is significant. There are 242 total liver-enriched genes. (**B**) The number of tissue-specific genes predicted from the hypergeometric distribution to be activated by HNF4A compared to the number actually activated. Liver- (p<10⁻⁸) and intestine enrichment (p<10⁻¹⁵) are significant. There are 242 total liver-enriched genes and 122 total intestine-enriched genes. (**C**) 242 liver genes characterized as activated by Foxa1, HNF4A, both, or neither. (**D**) 122 intestine genes characterized as activated by FOXA1, HNF4A, both, or neither.

occurred at the same sites bound in HepG2 liver cells (***Figure 3—figure supplement 1***). We conclude that FOXA1 and HNF4A have roughly equivalent abilities to bind and open inaccessible sites.

We sought to reconcile these findings with what the PFH had predicted. We first considered whether, in the absence of FOXA1, native K562 TFs were 'pioneering' for HNF4A. A de novo motif discovery analysis of the 500 bp centered on inaccessible FOXA1 or HNF4A binding sites revealed strong enrichment for each TF's motif, but no other strong signals. Similarly, we found no evidence for enrichment of predicted K562 PFs AP1 (FOS/JUN; MA0099.2; ***Biddie et al., 2011***), GATA1 (MA0035.4; ***Iwafuchi-Doi and Zaret, 2014***), MYB (MA0100.1; ***Lemma et al., 2021***), or SPI1 (PU.1; MA0080.1; ***Iwafuchi-Doi and Zaret, 2014***) either in inaccessible binding sites over randomly chosen sites or in HNF4A over FOXA1 binding sites (***Figure 3—figure supplement 2***). Thus, the similar activities of FOXA1 and HNF4A are not explained by pioneering activity provided by endogenous K562 TFs.

We next considered whether differences in FOXA1 and HNF4A motif content could explain our results. We focused on binding sites surrounding activated liver genes and used FOXA1 and HNF4A position weight matrices (***Figure 3G***) to count occurrences in the 500 bp of sequence surrounding these sites. Sites independently pioneered by FOXA1 contained between 2–4 motifs, while sites pioneered by HNF4A contained 3–6 motifs (***Figure 3H***). This is despite the fact that the FOXA1 motif

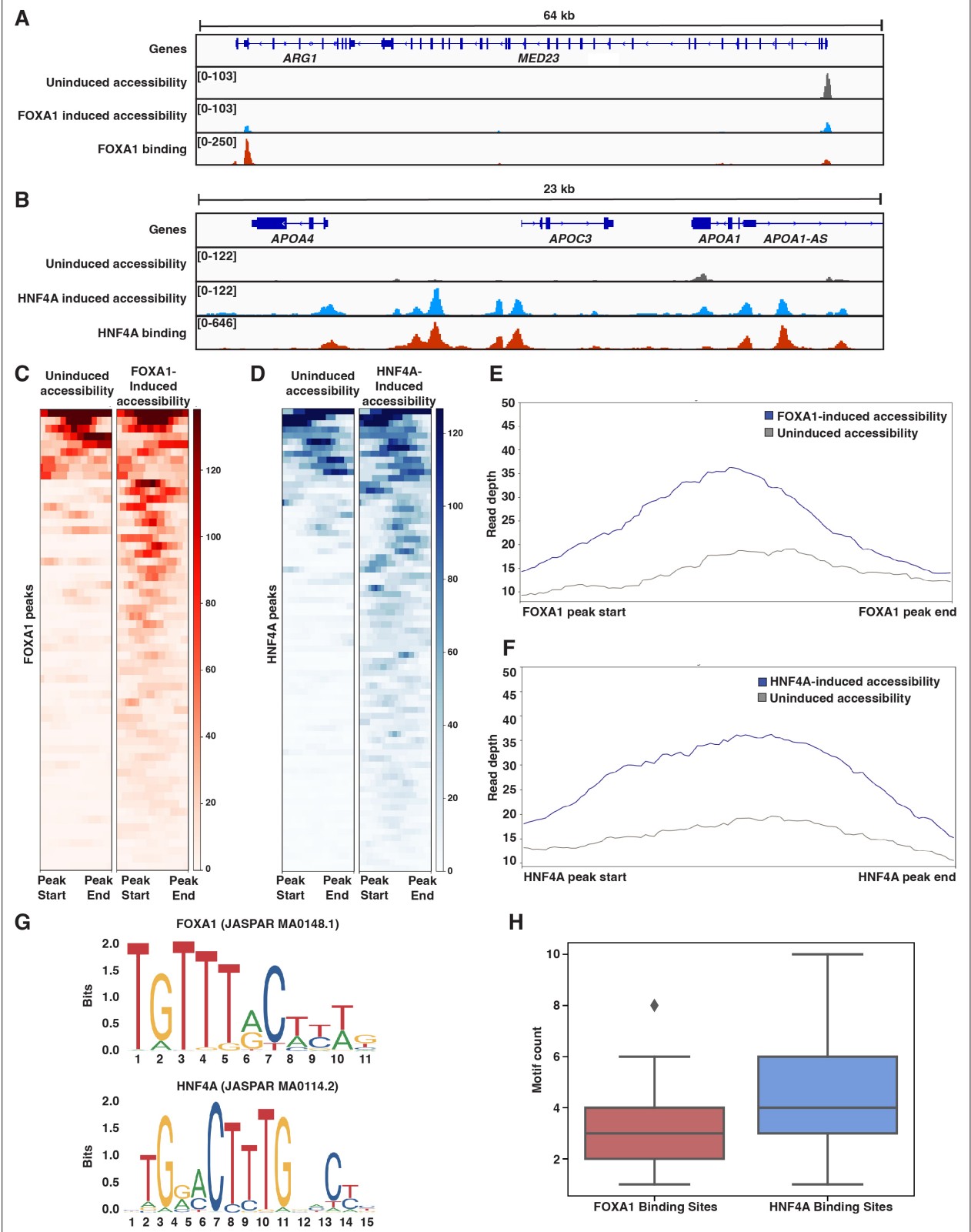

**Figure 3.** Both FOXA1 and HNF4A can pioneer liver-specific loci. (**A**) Genome browser view of a representative liver-specific locus (*ARG1*) in FOXA1 clonal line showing uninduced and induced accessibility and FOXA1 binding. (**B**) Genome browser view of a representative liver-specific locus (*APOC3*) in HNF4A clonal line showing uninduced and induced accessibility and HNF4A binding. (**C**) Heatmap of uninduced and induced accessibility at all FOXA1 binding sites within 50 kb of each FOXA1-activated liver-specific genes (n = 59). (**D**) Heatmap of uninduced and induced accessibility at all

*Figure 3 continued on next page*

*Figure 3 continued*

HNF4A binding sites within 50 kb of each HNF4A-activated liver-specific genes (n = 76). (**E**) Meta plot showing average signal across each site from (**C**). (**F**) Meta plot showing average signal across each site from (**D**). (**G**) Human FOXA1 and HNF4A sequence logo from JASPAR. (**H**) FOXA1 or HNF4A motif count within 500 bp centered upon FOXA1 or HNF4A binding sites within 50 kb of each FOXA1- or HNF4A-activated liver-specific genes, respectively. Motifs were called with FIMO using 1e-3 a p-value threshold. For each boxplot, the center line represents the median, the box represents the first to third quartiles, and the whiskers represent any points within 1.5× the interquartile range.

The online version of this article includes the following figure supplement(s) for figure 3:

**Figure supplement 1.** Characterization of FOXA1 and HNF4A binding patterns in FOXA1 or HNF4A individual clones.

**Figure supplement 2.** K562 transcription factor (TF) motif content in binding sites.

**Figure supplement 3.** FOXA1 and HNF4A motif scanning.

**Figure supplement 4.** Expression and binding at lower doxycycline induction.

occurs more frequently across the genome than the HNF4A motif (*Figure 3—figure supplement 3*). This observation is consistent with data showing that FOXA1 has higher affinity for its binding site than HNF4A (*Fernandez Garcia et al., 2019*; *Rufibach et al., 2006*; *Jiang et al., 1997*) and suggests that there may not be anything categorically different about FOXA1 and HNF4A, but rather that 'pioneer activity' may depend on the affinity of interaction between TF and DNA.

Another possible explanation for our results could be that at the concentrations TFs are expressed in cellular reprogramming, the differences between PFs and non-PFs are no longer apparent. We took advantage of our doxycycline-inducible system to test this hypothesis by lowering the doxycycline concentration from 0.5 µg/ml to 0.05 µg/ml, thus dropping the TF concentration significantly (*Figure 1—figure supplement 1*). We then remeasured binding and expression. We found that lower induction resulted in far fewer FOXA1 and HNF4A genome-wide binding events (*Figure 3—figure supplement 4*). This effect was even more pronounced when we subset the binding events into sites that were either accessible or inaccessible prior to induction. Both FOXA1 and HNF4A shifted from binding predominantly inaccessible sites to binding predominantly accessible sites (*Figure 3—figure supplement 4*). Thus, binding of both factors depends on a balance of TF concentration and accessibility state, and the results from expression profiling in the lower induction regime are consistent with this idea. Whereas FOXA1 and HNF4A previously activated 33 and 47 liver genes, at the lower induction rate they activated 8 and 30, respectively (*Figure 3—figure supplement 4*). Thus, lowering the induction levels had strong effects on the activities of both FOXA1 and HNF4A, but did not reveal qualitative differences between the two TFs. These results suggest that the induction conditions in cellular reprogramming do not mask differences between the TFs, a result consistent with the fact that the PFH was developed to explain the properties of cellular reprogramming cocktails.

## Some liver genes require cooperative FOXA1-HNF4A activity

In addition to those genes independently activated by FOXA1 and HNF4A, there is an additional set of 31 liver genes that are not activated until both FOXA1 and HNF4A are present (*Figure 4A*). We therefore asked whether these 31 liver genes are activated sequentially, as predicted by the PFH. If these genes conform to the PFH, then we would expect that at every gene there are nearby sites where FOXA1 binds individually and where FOXA1 and HNF4A co-bind when expressed together. This would be evidence for FOXA1 'pioneering' sites for later HNF4A binding and so we have called these sites 'FOXA1 pioneered' (FP). Sites are 'HNF4A pioneered' (HP) if HNF4A binds individually and FOXA1 and HNF4A co-bind when expressed together and sites are 'cooperatively bound' (CB) if neither TF binds individually but both do when expressed together.

When there is sequential binding of the two TFs it is apparent in comparisons of the single versus double expression clones, whereas obligate cooperativity between the TFs results in binding that is observed only in the double expression clone. There are examples of each modality surrounding *AMDHD1*, a liver-specific gene co-activated by FOXA1 and HNF4A (*Figure 4B*). When we examine all of the liver genes only activated by FOXA1-HNF4A co-expression, we find that in contradiction with the PFH, there are roughly equal numbers of FP, HP, and CB sites (*Figure 4C*). Therefore, in most cases, genes that require joint FOXA1-HNF4A activity do not rely on sequential FOXA1-then-HNF4A behavior.

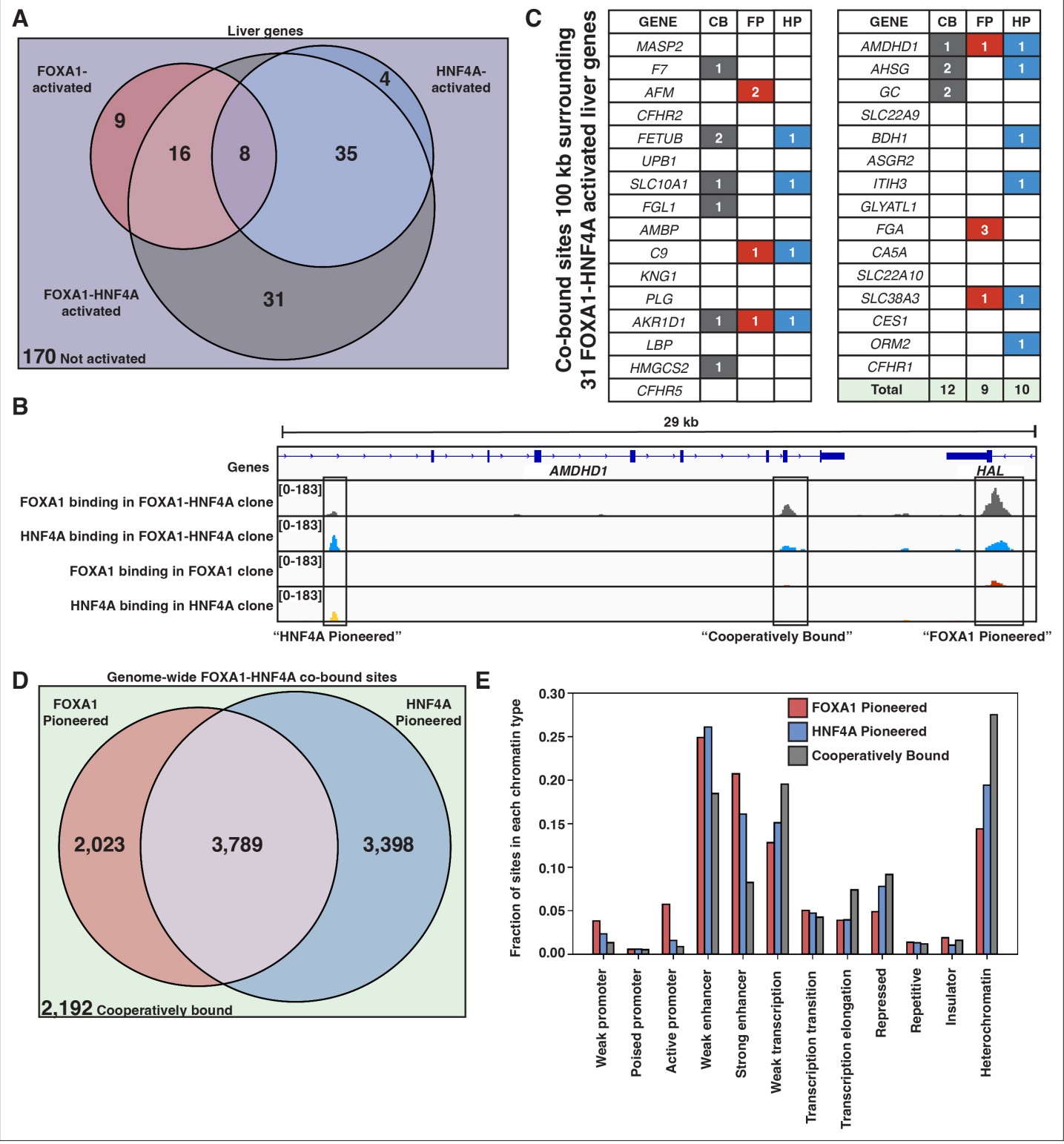

**Figure 4.** FOXA1 and HNF4A both pioneer and cooperative at liver-specific sites. (**A**) Venn diagram of all liver genes categorized as either activated by FOXA1, HNF4A, FOXA1-HNF4A, some combination, or by none of the three cocktails. (**B**) Genome browser view of a representative liver-specific locus (*AMDHD1*) showing examples of a co-bound site that is 'FOXA1 pioneered' (FP), 'HNF4A pioneered' (HP), and 'cooperatively bound' (CB). The first two tracks are FOXA1 and HNF4A binding in the FOXA1-HNF4A co-expression clone, and the last two tracks are FOXA1 and HNF4A binding in their individual expression clones. (**C**) List of the 31 liver genes that are only activated by FOXA1-HNF4A co-expression. The columns indicate how many co-bound FP, HP, or CB peaks exist within 100 kb of the gene. (**D**) Venn diagram of all genome-wide co-bound peaks categorized as either bound by

*Figure 4 continued on next page*

*Figure 4 continued*

FOXA1 individually (FP), HNF4A individually (HP), by both, or by neither (CB). (**E**) Overlap of FP, HP, and CB sites from (**D**) with ChromHMM annotations showing the fraction of each co-binding site type in each chromatin region.

The online version of this article includes the following figure supplement(s) for figure 4:

**Figure supplement 1.** Characterization of FOXA1-HNF4A differential accessibility.

The patterns of genome-wide co-binding and accessibility of FOXA1 and HNF4A follow similar trends. Of the 11,402 co-bound sites, 2023 were FP, 3398 were HP, and 2192 were CB (*Figure 4D*) and FOXA1-induced differentially accessible peaks explain a minority of the FOXA1-HNF4A differentially accessible peaks (*Figure 4—figure supplement 1*). Cooperative binding may be more important in less accessible parts of the region as there are more CB sites in ChromHMM-labeled (*Ernst and Kellis, 2012*) heterochromatic and repressed regions, and there are more FP and HP sites in promoter and enhancer regions (*Figure 4E*).

## Genome-wide motif analysis supports affinity model

The correlation between TF binding and factors such as TF binding strength, motif content, TF concentration, and accessibility state has so far suggested that an affinity model may explain ectopic FOXA1 and HNF4A behavior. Thus, we predicted that motif counts would explain genome-wide binding patterns. Because it requires more energy to bind at inaccessible sites than accessible sites, we predicted that there would be more motifs at inaccessible binding sites than at accessible sites, and that this motif distribution would be higher than that found in random genomic sequences. When we examined the 500 bp of sequence centered upon genome-wide TF binding sites, we found that for both FOXA1 and HNF4A, inaccessible binding sites had higher motif content than accessible binding sites and that these inaccessible binding sites had higher motif content than random inaccessible regions (*Figure 5A and B*). A simple motif threshold could predict binding, though only when predicting inaccessible sites (*Figure 5C*).

We also predicted that if FOXA1 and HNF4A are not categorically different, then we would find similar trends between the motifs for the two TFs. We predicted that total FOXA1 and HNF4A motif count at inaccessible sites would be higher than at random sites, and that FP or HP sites would have more FOXA1 or HNF4A sites, respectively, than CB sites. When we examined the 500 bp of sequence centered upon genome-wide co-bound sites, we found that there was higher total motif content at inaccessible binding sites as compared to random (*Figure 5D*) and that FOXA1 and HNF4A motif content was higher at FP or HP sites, respectively, than CB sites (*Figure 5E*). And like individually bound sites, a motif threshold could only predict inaccessible binding behavior (*Figure 5F*, top panels). The motif threshold was somewhat effective at differentiating between FP or HP versus CB sites (*Figure 5F*, lower panel). Altogether, these results further support our hypothesis that affinity better explains ectopic FOXA1 and HNF4a 'pioneer activity' than the current formulation of the PFH.

## Discussion

In contrast to the predictions of the PFH, we found that both the canonical PF FOXA1 and non-PF HNF4A can independently bind inaccessible sites, increase accessibility, and activate nearby endodermal genes in a mesodermal cell line. Some endodermal genes require the joint activity of both TFs, but the predominant mode of action at these targets does not conform to the predicted sequential activity of FOXA1 followed by HNF4A. These observations suggest that we do not need to invoke the PFH to explain FOXA1 and HNF4A's behavior in ectopic K562 cells and that instead we may use the affinity of interaction between each TF and its target sites to explain its behavior.

An affinity model assumes that there is nothing categorically different between FOXA1 and HNF4A. We hypothesize that differences still exist between TFs' abilities to bind at nucleosome-occluded sites but that 'pioneer activity' is a spectrum not a binary classifier. The probability of a binding event depends on the intrinsic binding ability of the TF and the motif count at a potential binding site. Previous measures of intrinsic binding strength that show FOXA1 binds more tightly than HNF4A (*Fernandez Garcia et al., 2019*; *Rufibach et al., 2006*; *Jiang et al., 1997*) may explain why in our assays FOXA1 requires fewer copies of its motif to bind. In fact, FOXA1 has a three-dimensional, histone-like structure that may explain its superior binding strength (*Clark et al., 1993*).

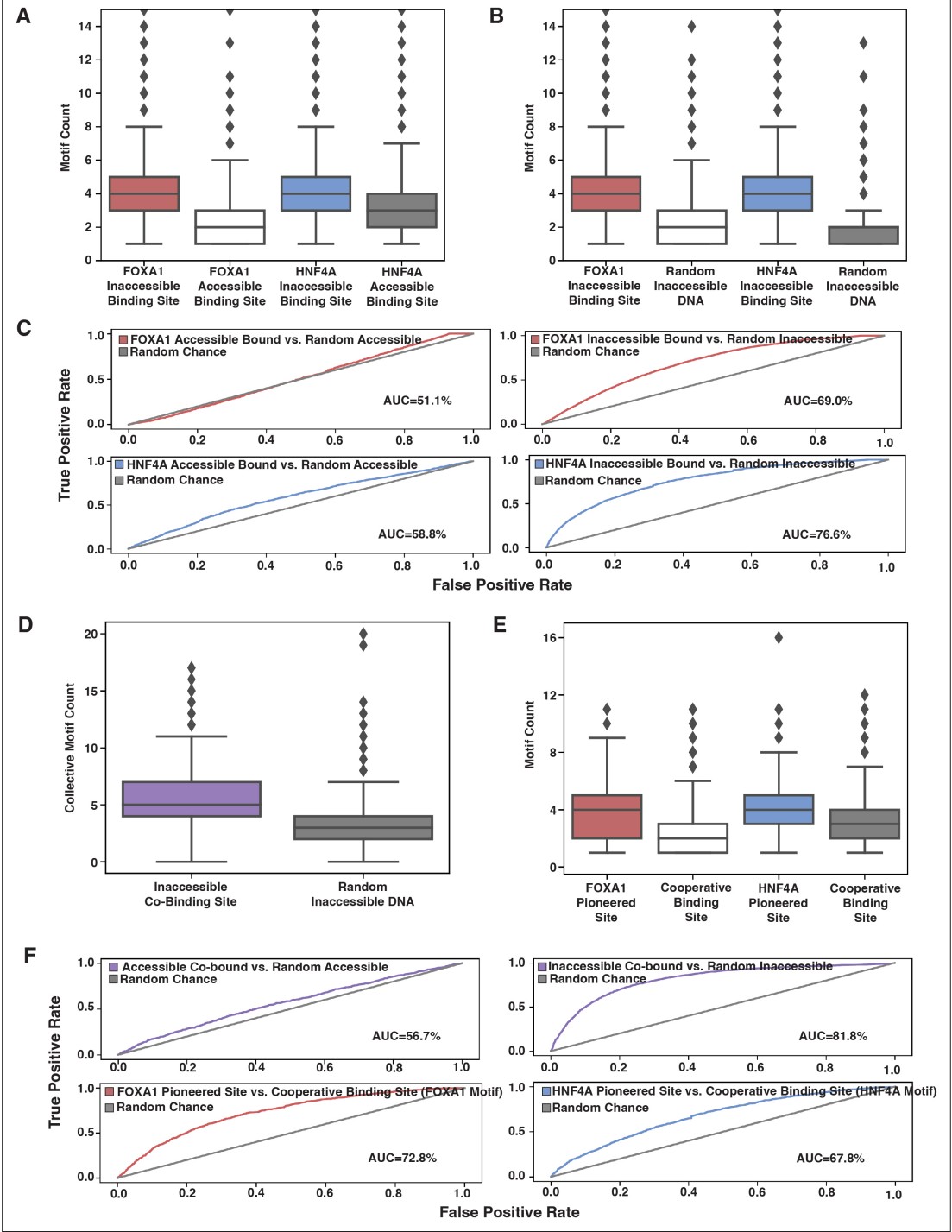

**Figure 5.** Affinity model predicts binding events. (**A**) FOXA1 or HNF4A motif count at all genomic occurrences of the respective transcription factor's (TF's) accessible or inaccessible binding sites. (**B**) FOXA1 or HNF4A motif count in genome-wide inaccessible binding sites versus length-matched random inaccessible DNA sequences. (**C**) Receiver operating characteristic (ROC) curves for predictive power of using sequence motif content to predict accessible (left panels) or inaccessible (right panels) binding sites from random sequence. (**D**) Total FOXA1 and HNF4A motif count at all genomic occurrences of inaccessible co-binding versus length-matched random inaccessible DNA sequences. (**E**) FOXA1 or HNF4A motif count in respective FOXA1 or HNF4A pioneered sites versus in cooperative binding sites (where neither TF bound individually). (**F**) ROC curves for predictive power of using sequence motif content to predict accessible or inaccessible co-binding events from random sequence (top panels) or to predict FOXA1 or HNF4A pioneered events from cooperative binding events. All FIMO scans used 1e-3 as p-value threshold and were conducted on 500 bp of sequence centered upon the binding site.

However, given the right sequence context, HNF4A also displays pioneer activity. We hypothesize that HNF4A was misclassified because of both developmental timing and indirect assays of pioneer activity. FOXA1 precedes HNF4A during hepatic development (*Lau et al., 2018*) and studies have traditionally established PF status by using endogenous binding or genome-wide chromatin marks. Perhaps sequential activity of FOXA1 and HNF4A is necessary during hepatic development, but our data show that both TFs are sufficient to independently activate silent genes.

We further hypothesize that our findings may extend to other reprogramming cocktails that combine PFs and non-PFs. While our study is limited to two TFs at two concentrations in one cell line, other data support our hypothesis. Early reprogramming of fibroblasts to myoblasts relied solely upon the ectopic overexpression of MyoD, without an accompanying non-PF (*Davis et al., 1987*; *Choi et al., 1990*) and new reprogramming cocktails have been tested and validated in a large-scale screen for single, cell-autonomous reprogramming TFs (*Ng et al., 2021*). Increasing the efficiency of reprogramming cocktails that depend on multiple TFs will require distinguishing between the independent and cooperative effects of TFs. For example, our finding that HNF4A independently activates more intestine-specific genes than FOXA1 raises the possibility that titrating down HNF4A activity during reprogramming could result in a more liver-specific profile. Such fine-tuning of TF activities has been suggested as an option to improve the success of other reprogramming cocktails (*Ma et al., 2015*; *Wang et al., 2015*; *Vaseghi et al., 2016*).

Although we found clear instances of sites independently pioneered by either FOXA1 or HNF4A, not all sites containing multiple motifs were pioneered in K562 cells, which comports with studies showing that the sequence context in which motifs occur also plays an important role in determining whether sites will be pioneered or not. GAL4's ability to bind nucleosomal DNA templates depends both on the number of copies of its motif (*Taylor et al., 1991*) and the positioning of the motif in the nucleosome (*Vettese-Dadey et al., 1994*). Precise nucleosome positioning also dictates TP53 and OCT4 pioneering behavior (*Yu and Buck, 2019*; *Huertas et al., 2020*). A TF's motif affinity, motif count, and the presence of cofactor motifs are all strong predictors of pioneer activity (*Yan et al., 2018*; *Manandhar et al., 2017*; *Donaghey et al., 2018*; *Heinz et al., 2010*; *Boyes and Felsenfeld, 1996*; *Minderjahn et al., 2020*; *Meers et al., 2019*) and certain types of heterochromatic patterning have been labeled 'pioneer resistant' (*Mayran et al., 2018*). Thus, we hypothesize that general pioneer activity may best be summarized by the free energy balance between TFs, nucleosomes, and DNA (*Polach and Widom, 1996*; *Mirny, 2010*) rather than as a property of specific classes of TFs.

## Materials and methods

### Key resources table

| Reagent type (species) or resource | Designation | Source or reference | Identifiers | Additional information |
|---|---|---|---|---|
| Strain, strain background (*Homo sapiens*) | FOXA1 | K562 | Cat# CCL-243 (ATCC); RRID:CVCL_0004 | Infected with pINDUCER21 lentiviral vector (*Meerbrey et al., 2011*) (Addgene#46948) carrying FOXA1 ORF (Addgene#120438) |
| Strain, strain background (*H. sapiens*) | HNF4A | K562 | Cat# CCL-243 (ATCC); RRID:CVCL_0004 | Infected with pINDUCER21 lentiviral vector (*Meerbrey et al., 2011*) (Addgene#46948) carrying HNF4A ORF (Addgene#120450) |
| Strain, strain background (*H. sapiens*) | FOXA1-HNF4A | K562 | Cat# CCL-243 (ATCC); RRID:CVCL_0004 | Infected with pINDUCER21 lentiviral vector (*Meerbrey et al., 2011*) (Addgene#46948) carrying FOXA1 ORF and then HNF4A ORF |
| Chemical compound, drug | Doxycycline | Sigma | Cat# D9891-1G | Treated at 0.5 and 0.05 µg/ml |
| Chemical compound, drug | Polybrene | Sigma | Cat# TR1003G | Treated at 10 µg/ml |
| Commercial assay or kit | PureLink RNA Mini | Invitrogen | Cat# 12183020 | |
| Commercial assay or kit | PureLink DNase | Invitrogen | Cat# 12185010 | |
| Commercial assay or kit | ReadyScript cDNA Synthesis Mix | Sigma | Cat# RDRT-100RXN | |

*Continued on next page*

*Continued*

| Reagent type (species) or resource | Designation | Source or reference | Identifiers | Additional information |
|---|---|---|---|---|
| Commercial assay or kit | SYBR Green PCR Master Mix | Applied Biosystems | Cat# 4301955 | |
| Commercial assay or kit | NEBNext Ultra II Directional RNA Library Prep Kit | NEB | Cat# E7765S | |
| Commercial assay or kit | AMPure XP beads | Beckman Coulter | Cat# A63880 | |
| Commercial assay or kit | pAG-TN5 | EpiCypher | Cat# 15-1017 | |
| Commercial assay or kit | Concanavalin A paramagnetic beads | EpiCypher | Cat# 21-1401 | |
| Commercial assay or kit | HiFi DNA assembly | NEB | Cat# E2621L | |
| Antibody | Anti-FOXA1 (Rabbit monoclonal) | Cell Signaling | Cat# 53528; RRID:AB_2799438 | (1:100) |
| Antibody | Anti-HNF4A (mouse monoclonal) | Invitrogen | Cat# MA1-199; RRID:AB_2633309 | (1:100) |
| Antibody | Anti-H3K4me3 (Rabbit polyclonal) | EpiCypher | Cat# 13-0041 | (1:50) |
| Antibody | Anti-rabbit (goat polyclonal) | EpiCypher | Cat# 13-0047 | (1:100) |
| Antibody | Anti-mouse (goat polyclonal) | EpiCypher | Cat# 13-0048 | (1:100) |
| Software, algorithm | Salmon | https://combine-lab.github.io/salmon/getting_started/ | https://doi.org/10.1038/nmeth.4197; RRID:SCR_017036 | |
| Software, algorithm | DESeq2 | https://bioconductor.org/packages/release/bioc/html/DESeq2.html | https://doi.org/10.1186/s13059-014-0550-8; RRID:SCR_015687 | |
| Software, algorithm | deepTools2 | https://deeptools.readthedocs.io/en/develop/ | https://doi.org/10.1093/nar/gkw257; RRID:SCR_016366 | |
| Software, algorithm | bowtie2 | http://bowtie-bio.sourceforge.net/bowtie2/index.shtml | https://doi.org/10.1038/nmeth.1923; RRID:SCR_016368 | |
| Software, algorithm | MACS2 | https://pypi.org/project/MACS2/ | https://doi.org/10.1186/gb-2008-9-9-r137; RRID:SCR_013291 | |
| Software, algorithm | featureCounts | https://www.rdocumentation.org/packages/Rsubread/versions/1.22.2/topics/featureCounts | https://doi.org/10.1093/bioinformatics/btt656; RRID:SCR_012919 | |
| Software, algorithm | IDR | https://www.encodeproject.org/software/idr/ | https://doi.org/10.1214/11-AOAS466; RRID:SCR_017237 | |
| Software, algorithm | DiffBind | https://bioconductor.org/packages/release/bioc/html/DiffBind.html | https://doi.org/10.18129/B9.bioc.DiffBind; RRID:SCR_012918 | |
| Software, algorithm | BEDTools | https://bedtools.readthedocs.io/en/latest/ | https://doi.org/10.1093/bioinformatics/btq033; RRID:SCR_006646 | |
| Software, algorithm | STREME | https://meme-suite.org/meme/tools/streme | https://doi.org/10.1093/bioinformatics/btab203; RRID:SCR_001783 | |
| Software, algorithm | FIMO | https://meme-suite.org/meme/tools/fimo | https://doi.org/10.1093/bioinformatics/btr064; RRID:SCR_001783 | |

## Cell lines

We grew K562 cells (ATCC CCL-243, Manassas, VA) in Iscove's Modified Dulbecco Serum supplemented with 10% fetal bovine serum, 1% penicillin-streptomycin, and 1% nonessential amino acids. We used these cells to generate our clonal lines (FOXA1, HNF4A, and FOXA1-HNF4A), and we thank Washington University in St. Louis Genome Engineering and the iPSC Center for their help confirming K562 identity with STR profiling and testing for mycoplasma contamination. When it was

time to conduct one of our functional assays, we split FOXA1-, HNF4A-, or FOXA1-HNF4A-expressing cells into replicate flasks and then treated with either ±0.5 µg/ml or 0.05 µg/ml doxycycline (Sigma #D9891-1G) for 24 hr.

## Cloning, production, and infection of viral vectors

We used PCR to add V5 epitope tags to the 3′ end of FOXA1 (Addgene #120438, Watertown, MA) and HNF4A (Addgene #120450) constructs and then used HiFi DNA Assembly (NEB #E2621L, Ipswich, MA) to clone each construct into a pINDUCER21 doxycycline-inducible lentiviral vector (Addgene #46948). All primers are listed in *Supplementary file 3*. The Hope Center Viral Vector Core at Washington University in St. Louis then generated and titered high-concentration virus. We infected human K562 cells at a MOI of 1 by spinoculation at 800G for 30 min in the presence of 10 µg/ml polybrene (Sigma #TR1003G, St. Louis, MO), passaged the cells for 3 days, and then selected for positively infected cells by single-cell sorting on GFP+ into 96-well plates. Finally, we used qPCR to select for clones that had high inducibility of TF and target gene expression (*Figure 1—figure supplement 1*).

## RNA extractions, reverse transcription, and qPCR

We extracted RNA from 1e6 cells/sample with the PureLink RNA Mini (Invitrogen #12183020, Waltham, MA) column extraction kit and completed on-column DNA digestion with PureLink DNase (Invitrogen #12185010). We quantified and assessed the quality of the RNA with an Agilent 2200 Tapestation instrument and then either froze down pure RNA for later RNA-sequencing library preparation or used ReadyScript cDNA Synthesis Mix (Sigma #RDRT-100RXN) to produce cDNA for qPCR. We performed qPCR with SYBR Green PCR Master Mix (Applied Biosystems #4301955, Waltham, MA) and gene-specific and housekeeping primers (*Supplementary file 3*).

## RNA-sequencing and analysis

We generated three replicates of ±doxycycline-treated RNA-sequencing libraries with the NEBNext Ultra II Directional RNA Library Prep Kit (NEB #E7765S). We quantified and assessed the quality of the libraries with an Agilent 2200 Tapestation instrument, size selected with AMPure XP beads (Beckman Coulter #A63880, Brea, CA), and then sequenced the libraries with 75 bp paired-end reads on an Illumina NextSeq 500 instrument.

We quantified transcripts with Salmon (*Patro et al., 2017*), filtered out any with fewer than 10 reads, and then called differentially expressed transcripts with DESeq2 (*Love et al., 2014*). A gene was called differentially upregulated if it had a log2fold change of at least 1 and was called 'activated' if it had fewer than 50 normalized reads in the uninduced control. A gene was called 'tissue-specific' according to the Human Protein Atlas definition of tissue enrichment (*Uhlén et al., 2015*), which is if a gene is at least fourfold higher expressed in the tissue of interest than in any other tissue as measured by deep sequencing of RNA from the tissue of interest.

## ATAC-sequencing and analysis

We followed the Omni-ATAC protocol (*Corces et al., 2017*) to generate two replicates of ±doxycycline-treated low-background ATAC-sequencing libraries. We isolated 2e5 cells/sample and then extracted 5e4 nuclei/sample for tagmentation and library preparation. We quantified and assessed the quality of the libraries with an Agilent 2200 Tapestation instrument, size selected with AMPure XP beads, and then sequenced the libraries with 75 bp paired-end reads on an Illumina NextSeq 500 instrument.

We aligned transcripts with bowtie2 (*Langmead and Salzberg, 2012*) with the parameters: `--local -X2000`, generated RPKM normalized BigWig files for visualization with deepTools bamCoverage (*Ramírez et al., 2016*), and then called peaks at low stringency with MACS2 (p=0.01) (*Zhang et al., 2008*). With these peaks, we either called reproducible peaks with IDR (FDR of 0.05) (*Li et al., 2011*) or used DiffBind (*Stark and Brown, 2011*) to call differential peaks. We calculated the Fraction of Reads in Peaks (FRiP) with the Subread featureCounts tool (*Liao et al., 2014*), counting reads for each replicate in the IDR-merged peak list (*Supplementary file 4*).

## CUT&Tag and analysis

We followed the CUTANA Direct-to-PCR CUT&Tag protocol (EpiCypher, Chapel Hill, NC) to generate two replicates of low-background CUT&Tag libraries. We isolated 1e5 cells/sample, extracted nuclei

with Concanavalin A paramagnetic beads (EpiCypher #21-1401), and then either used rabbit anti-human FOXA1 monoclonal antibody (Cell Signaling #53528, Danvers, MA), mouse anti-human HNF4A monoclonal antibody (Invitrogen #MA1-199), or rabbit anti-human histone H3K4me3 polyclonal antibody (EpiCypher #13-0041) as a positive control. We amplified this signal with either goat anti-rabbit (EpiCypher #13-0047) or goat anti-mouse (EpiCypher #13-0048) polyclonal secondary antibodies. For a negative control, we omitted the primary antibody and checked for any nonspecific pull-down. Finally, we used CUTANA pAG-Tn5 (EpiCypher #15-1017) to tagment the genomic regions surrounding each bound antibody complex. We quantified and assessed the quality of the libraries with an Agilent 2200 Tapestation instrument, size selected with AMPure XP beads, and then sequenced the libraries with 150 bp paired-end reads on an Illumina NextSeq 500 instrument.

When we assessed our libraries with the Agilent Tapestation instrument, we found that our negative controls had minimal signal. This is expected in the protocol, and as such sequencing the sample is recommended as optional (*Kaya-Okur et al., 2020*). For this reason, we sequenced only our positive samples. We aligned our samples with bowtie2 (*Langmead and Salzberg, 2012*) using recommended parameters (*Kaya-Okur et al., 2020*): `--very-sensitive --end-to-end --no-mixed --no-discordant -I 10X700`, created RPKM normalized BigWig files with deepTools bamCoverage (*Ramírez et al., 2016*), and called peaks with MACS2 (p=1e-5) (*Zhang et al., 2008*) with recommended parameters (*Kaya-Okur et al., 2019*). We calculated the FRiP with Subread featureCounts tool (*Liao et al., 2014*; *Supplementary file 5*). We then combined overlapping peaks from replicate samples using BEDTools intersect (*Quinlan and Hall, 2010*). We attributed binding sites to genes if they were within 50 kb (25 kb up- and 25 kb downstream) of the gene's TSS. Because co-binding occurred less frequently, we attributed co-binding sites to genes if they were within 100 kb of the gene's TSS. 'FOXA1 pioneered' sites were those where we identified overlapping FOXA1 and HNF4A binding peaks within 100 kb of a gene that was only activated by FOXA1 and HNF4A and where there was also an overlapping FOXA1 binding peak, when FOXA1 was expressed alone. 'HNF4A pioneered' sites were those where we identified overlapping FOXA1 and HNF4A binding peaks within 100 kb of a gene that was only activated by FOXA1 and HNF4A and where was also an overlapping HNF4A binding peak, when HNF4A was expressed alone. And 'cooperatively bound' sites were those where we identified overlapping FOXA1 and HNF4A binding peaks within 100 kb of a gene that was only activated by FOXA1 and HNF4A and where there was neither a FOXA1 nor HNF4A binding peak.

## Tissue- and biological process-specific expression analysis

We generated lists of tissue-specific genes for each tissue by extracting 'enriched genes' from the Human Protein Atlas, as detailed above. We then computed hypergeometric assays to determine if our activated genes were enriched in any tissue-specific gene set. Finally, we used Panther gene ontology analysis to identify enriched biological processes.

## Genome tracks and profile plot analysis

We visualized the signal from our functional assays by loading each file into the Integrated Genome Viewer (*Robinson et al., 2011*) using hg19 as reference. We then used the computeMatrix function in reference point mode and plotProfile function, both with default parameters, in the deepTools suite (*Ramírez et al., 2016*) to display aggregated CUT&Tag and ATAC-sequencing signals across indicated genomic regions.

## Motif and chromatin segmentation analysis

Before running motif scans, we extracted 500 bp of sequence centered on the binding sites of interest. Then, we used STREME (*Bailey, 2021*) for de novo motif discovery and FIMO (*Grant et al., 2011*) for specific motif occurrence counting. We used 1e-3 as a p-value threshold and JASPAR (*Fornes et al., 2020*) PWMs for FOXA1 (MA0148.1) and HNF4A (MA0114.2). To use motif content to predict binding, we lowered the p-value threshold to 0 to allow for weak motif contributions and then summed the motif content for each sequence. A simple threshold on this aggregate score was used as a classifier, with the receiver operating characteristic (ROC) curves generated by sweeping this threshold and plotting the resulting true-positive rates against false-positive rates. We used ChromHMM annotations (*Ernst and Kellis, 2012*) to characterize the epigenetic profile of FOXA1 and HNF4A binding sites.

## Acknowledgements

We thank Dr. Gary Stormo, Dr. Robi Mitra, and members of the Cohen Lab for reading and critiquing the manuscript and for helpful discussion; Jessica Hoisington-Lopez and MariaLynn Crosby in the DNA Sequencing Innovation Lab for assistance with high-throughput sequencing; the Genome Engineering and iPSC Center for allowing us to use their Sony Flow Cytometer for cell sorting; and Mingjie Li in the Hope Center Viral Vectors Core for assistance with producing lentiviral expression vectors.

## Additional information

### Funding

| Funder | Grant reference number | Author |
|---|---|---|
| National Institute of General Medical Sciences | R01GM092910 | Barak A Cohen |
| National Human Genome Research Institute | T32HG000045 | Barak A Cohen |
| National Institute of General Medical Sciences | T32GM007200 | Jeffrey L Hansen |

The funders had no role in study design, data collection and interpretation, or the decision to submit the work for publication.

### Author contributions

Jeffrey L Hansen, Conceptualization, Data curation, Formal analysis, Investigation, Methodology, Software, Validation, Visualization, Writing – original draft, Writing – review and editing; Kaiser J Loell, Formal analysis, Visualization; Barak A Cohen, Conceptualization, Funding acquisition, Project administration, Supervision, Writing – original draft, Writing – review and editing

### Author ORCIDs

Jeffrey L Hansen http://orcid.org/0000-0001-5343-9066
Kaiser J Loell http://orcid.org/0000-0001-7076-6659
Barak A Cohen http://orcid.org/0000-0002-3350-2715

### Decision letter and Author response

Decision letter https://doi.org/10.7554/eLife.73358.sa1
Author response https://doi.org/10.7554/eLife.73358.sa2

## Additional files

### Supplementary files

• Supplementary file 1. FOXA1 gene ontology analysis. Gene ontology terms, representative genes, and FDR values for liver-specific genes activated by FOXA1.

• Supplementary file 2. HNF4A gene ontology analysis. Gene ontology terms, representative genes, and FDR values for liver-specific genes activated by HNF4A.

• Supplementary file 3. Primer sequences. Primer sequences used for plasmid construction and qPCR analysis.

• Supplementary file 4. ATAC-sequencing quality summary statistics. Sequencing statistics for each ATAC-sequencing run, including the read length, read count, peak count, and Fraction of Reads in Peaks (FRiP).

• Supplementary file 5. CUT&Tag sequencing quality summary statistics. Sequencing statistics for each CUT&Tag run, including the read length, read count, peak count, and Fraction of Reads in Peaks (FRiP).

• Transparent reporting form

## Data availability

All genomic sequencing data have been deposited on Gene Expression Omnibus (GEO) under accession number GSE182191.

The following dataset was generated:

| Author(s) | Year | Dataset title | Dataset URL | Database and Identifier |
|---|---|---|---|---|
| Hansen JL, Cohen BA | 2021 | A Test of the Pioneer Factor Hypothesis | https://www.ncbi.nlm.nih.gov/geo/query/acc.cgi?acc=GSE182191 | NCBI Gene Expression Omnibus, GSE182191 |

The following previously published dataset was used:

| Author(s) | Year | Dataset title | Dataset URL | Database and Identifier |
|---|---|---|---|---|
| Partridge EC, Chhetri SB, Prokop JW, Ramaker RC, Jansen CS, Goh ST, Mackiewicz M, Newberry KM, Brandsmeier LA, Meadows SK, Messer CL, Hardigan AA, Coppola CJ, Dean EC, Jiang S, Savic D, Mortazavi A, Wold BJ, Myers RM, Mendenhall EM | 2020 | Occupancy maps of 208 chromatin-associated proteins in one human cell type | https://www.ncbi.nlm.nih.gov/geo/query/acc.cgi?acc=GSE104247 | NCBI Gene Expression Omnibus, GSE104247 |

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
