## [Editor Report]

The article elegantly tests a leading hypothesis in the field and demonstrates that the pioneer factor model is not sufficient to explain most of the gene activation when liver transcription factors (TFs) are ectopically expressed. These results provide a new standard for testing the degree to which a TF acts as a pioneer factor. Moreover, they suggest that cis elements (i.e., motif density) are critical to open a locus upon transcription factor expression. How motif density and TF binding contribute to nucleosome eviction will be interesting to unravel in future studies.

---

## [Decision Letter]

**Decision letter after peer review:**

Thank you for submitting your article "A Test of the Pioneer Factor Hypothesis" for consideration by *eLife*. Your article has been reviewed by 2 peer reviewers, and the evaluation has been overseen by a Reviewing Editor and Jessica Tyler as the Senior Editor. The reviewers have opted to remain anonymous.

Essential revisions:

1) Tone down and clarify conclusions. See reviewer comments below for details. In particular, the title is misleading and should be edited to reflect the more narrow scope of the manuscript. Also, the limitations of the experimental system should be made clear. Finally, there should be more of an attempt to reconcile their findings with prior studies.

2) Various data analyses to address reviewer comments, specifically address the following ones:

A) Are the sites that are independently activated by FOXA1 or HNF4A bound by other TFs as well? Specifically, is it the case that FOXA1 and HNF4A are acting like solo pioneering factors at these sites, or are FOXA1 and HNF4A similarly "cooperatively co-bound" at these sites with TFs that are endogenously expressed in K562 cells but on their own do not open up these elements?

B) A more thorough analysis could substantially strengthen the manuscript. Most features in this manuscript are classified in a Boolean OFF/ON manner, which creates clean analysis, but limits the scope of interpretations. By oversimplifying the assumptions, this limits the ability to explain the results in the context of sequence affinity. Do Foxa1 and Hnf4a bind with equal amounts to co-bound accessible sites? Does their binding enrichment (characterized by CUT&Run) correlate with differential pre- and post- induction ATAC-seq enrichment? It is mentioned in the discussion that "not all sites containing multiple motifs were pioneered in K652 cells" and suggests many possible explanations such as (1) TF affinity to its motif, (2) sequence context, (3) motif orientation relative to the nucleosome. However, aside from running FIMO to scan for motif instances, there is limited follow-up analysis. At the very least, FIMO could be used to quantitatively score motif matches in the genome in order to explore the role of sequence affinity.

3) Determine how TF concentration impacts the results. The TF concentration very likely determines the degree by which chromatin is made accessible. Since the authors have the perfect experimental system to address this question, it would be straightforward to determine whether changing TF concentration influences the ability of Hnf4a to open chromatin comparably to Foxa1.

*Reviewer #2 (Recommendations for the authors):*

Please provide data on the level of expression of FOXA1 and HNF4A relative to their physiological expression levels in human tissue/cell lines.

The presentation of Supplementary Figure 2 and 3 is confusing. Is Supplementary Figure 3a stating that there are ~35,000 FoxA1 CUT&Tag peaks in uninduced cells? Or is this showing the ATAC-seq chromatin accessibility in uninduced cells for FoxA1 CUT&Tag peaks that were identified in induced cells?

Page 6, line 137. The line "The vast majority of both FoxA1 and Hnf4a binding sites fell within inaccessible regions…" is confusing, as it implies that the FoxA1 and Hnf4a CUT&Tag peaks were outside of ATAC-seq peaks in the same sample. I am assuming that you meant to say that the "The vast majority of both FoxA1 and Hnf4a binding sites in the induced sample matched inaccessible regions in the uninduced sample…"

Please provide quality summary statistics for ATAC-seq and CUT&Tag data. Were these samples sequenced to a similar degree? Do the ATAC-seq samples have a similar proportion of reads surrounding TSSs, or within peaks? Were a similar number of peaks called in each ATAC-seq sample?

Pg 12, line 235. "cooperative" is a better word than "collaborative" to describe this phenomenon. Similarly, the term "collaboratively co-bound" is better represented using the term "cooperatively co-bound". Note that the term cooperative does not necessitate direct protein-protein interactions between TFs, but can rather merely reflect their cooperative ability to stabilize a nucleosome free stretch of DNA (PMID 21149679).

Please provide further description on how you define the list of organ-specific genes?

Per convention, please use uppercase protein names as K562 is a human cell line.

Are the sites that are independently activated by FOXA1 or HNF4A bound by other TFs as well? Specifically, is it the case that FOXA1 and HNF4A are acting like solo pioneering factors at these sites, or are FOXA1 and HNF4A similarly "cooperatively co-bound" at these sites with TFs that are endogenously expressed in K562 cells but on their own do not open up these elements?

*Reviewer #3 (Recommendations for the authors):*

The key weakness in this manuscript is that the authors question the pioneer factor hypothesis based on a single finding. They find that Hnf4a, which was classified previously as non-pioneer, is able to open chromatin in their assay. Furthermore, they do not sufficiently explore the similarities and differences between the binding behavior of Foxa1 and Hnf4a to provide more insights. As it is, they do not refute the pioneer factor hypothesis and thus the title is misleading. There are several main points that would strengthen the manuscript:

1) If Hnf4a is also able to open chromatin, why has it previously been categorized as a non-pioneer? Is it because of its role as "differentiation TF", thus a TF that is expressed relatively late in development? Or are there physical properties of the TF that make it a non-pioneer? Some of the cited papers actually describe Hnf1a whose behavior is quite similar to pioneer TFs. Without exploring this, the simplest explanation for the observed results is that Hnf4a has previously been misclassified.

2) A more thorough analysis could substantially strengthen the manuscript. Most features in this manuscript are classified in a Boolean OFF/ON manner, which creates clean analysis, but limits the scope of interpretations. By oversimplifying the assumptions, this limits the ability to explain the results in the context of sequence affinity. Do Foxa1 and Hnf4a bind with equal amounts to co-bound accessible sites? Does their binding enrichment (characterized by CUT&Run) correlate with differential pre- and post- induction ATAC-seq enrichment? It is mentioned in the discussion that "not all sites containing multiple motifs were pioneered in K652 cells" and suggests many possible explanations such as (1) TF affinity to its motif, (2) sequence context, (3) motif orientation relative to the nucleosome. However, aside from running FIMO to scan for motif instances, there is limited follow-up analysis. At the very least, FIMO could be used to quantitatively score motif matches in the genome in order to explore the role of sequence affinity.

3) What is the effect of TF concentration? I understand the rationale of the experimental design to express Foxa1 and Hnf4a at similar levels in order to test their ability to open chromatin at the same concentration. However, the TF concentration very likely determines the degree by which chromatin is made accessible. Since the authors have the perfect experimental system to address this question, there is a missed opportunity to explore whether changing TF concentration influences the ability of Hnf4a to open chromatin comparably to Foxa1.

---

## [Author Response]

Essential revisions:1) Tone down and clarify conclusions. See reviewer comments below for details. In particular, the title is misleading and should be edited to reflect the more narrow scope of the manuscript.

We have changed the title and adjusted our language to reflect the focus of our study on FOXA1, HNF4A, and their role in inducing ectopic expression of liver-specific genes. We chose FOXA1 and HNF4A because they offer a clear and testable example of the PFH where a PF and nonPF have been hypothesized to act sequentially. We have also clarified that our results do not necessarily apply to all other combinations of PFs and nonPFs (lines 1, 35, 52, 82, 293, 341).

Also, the limitations of the experimental system should be made clear.

We have emphasized that our conclusions are based on the behavior of two factors in one cell line (line 317), but also have directly addressed reviewer comments regarding TF concentration and endogenous K562 co-factors (see below).

Finally, there should be more of an attempt to reconcile their findings with prior studies.

We have added a section that discusses why HNF4A may have been mis-classified as a nonPF (line 308).

2) Various data analyses to address reviewer comments, specifically addressthe following ones:A) Are the sites that are independently activated by FOXA1 or HNF4A bound by other TFs as well? Specifically, is it the case that FOXA1 and HNF4A are acting like solo pioneering factors at these sites, or are FOXA1 and HNF4A similarly "cooperatively co-bound" at these sites with TFs that are endogenously expressed in K562 cells but on their own do not open up these elements?

We have Figure 3—figure supplement 2 and accompanying text (line 191) that demonstrates that binding motifs for endogenous K562 factors are not more enriched at FOXA1 or HNF4a inaccessible binding sites than expected by random chance, nor are they differentially enriched between FOXA1 vs. HNF4A binding sites.

B) A more thorough analysis could substantially strengthen the manuscript. Most features in this manuscript are classified in a Boolean OFF/ON manner, which creates clean analysis, but limits the scope of interpretations. By oversimplifying the assumptions, this limits the ability to explain the results in the context of sequence affinity. Do Foxa1 and Hnf4a bind with equal amounts to co-bound accessible sites? Does their binding enrichment (characterized by CUT&Run) correlate with differential pre- and post- induction ATAC-seq enrichment? It is mentioned in the discussion that "not all sites containing multiple motifs were pioneered in K652 cells" and suggests many possible explanations such as (1) TF affinity to its motif, (2) sequence context, (3) motif orientation relative to the nucleosome. However, aside from running FIMO to scan for motif instances, there is limited follow-up analysis. At the very least, FIMO could be used to quantitatively score motif matches in the genome in order to explore the role of sequence affinity.

We have added Figure 5 and accompanying text to show that an affinity model more parsimoniously explains ectopic FOXA1 and HNF4A binding behavior than the Pioneer Factor Hypothesis (lines 262-286, lines 298-306, and line 470). We show that the total motif count distinguishes between binding sites and random DNA, between inaccessible binding sites and accessible binding sites, and between sites “pioneered” or “cooperatively bound” sites. In each case, we can equally well predict FOXA1 and HNF4A binding with motif count, suggesting that the two TFs are not behaving qualitatively differently.

3) Determine how TF concentration impacts the results. The TF concentration very likely determines the degree by which chromatin is made accessible. Since the authors have the perfect experimental system to address this question, it would be straightforward to determine whether changing TF concentration influences the ability of Hnf4a to open chromatin comparably to Foxa1.

In response to this comment we performed new experiments in which we reduced the levels of FOXA1 and HNF4A induction by lowering the doxycycline concentration from 0.5 to 0.05 ug/ml. The lower induction conditions reduce binding and liver gene activation for both FOXA1 and HNF4A, but does not qualitatively alter the results: the effects are still the same for both FOXA1 and HNF4A (lines 214-231 and Figure 3—figure supplement 4).

Reviewer #2 (Recommendations for the authors):Please provide data on the level of expression of FOXA1 and HNF4A relative to their physiological expression levels in human tissue/cell lines.

It is unclear what expression levels of FOXA1 and HNF4A are physiological, as their levels in HepG2 or even primary fibroblasts are not physiological. It was partly for this reason that we chose to study the PFH in the context of cellular reprogramming (where it was first conceived) and why we followed protocols used during reprogramming assays. Our concentrations may be higher than during development, but they are the same as the levels used in cellular reprogramming protocols. To address the concern that our results might be an artifact of overexpression, we lowered the induction conditions 10-fold and re-measured binding and ectopic expression for both FOXA1 and HNF4A. As discussed above, the lower induction conditions reduce binding and liver gene activation for both FOXA1 and HNF4A, but does not qualitatively alter the results: the effects are still the same for both FOXA1 and HNF4A. We also have taken careful steps to limit the degree of over-expression and to express FOXA1 and HNF4A at similar levels (line 109 and Figure 1-figure supplement 1).

The presentation of Supplementary Figure 2 and 3 is confusing. Is Supplementary Figure 3a stating that there are ~35,000 FoxA1 CUT&Tag peaks in uninduced cells? Or is this showing the ATAC-seq chromatin accessibility in uninduced cells for FoxA1 CUT&Tag peaks that were identified in induced cells?

We now clarify in the text that Figure 1—figure supplement 2 and Figure 3—figure supplement 1 show the chromatin accessibility in uninduced (-dox) cells for FOXA1 or HNF4A CUT&Tag peaks that were identified in induced (+dox) cells (lines 142, 185, 715, and 752).

Page 6, line 137. The line "The vast majority of both FoxA1 and Hnf4a binding sites fell within inaccessible regions…" is confusing, as it implies that the FoxA1 and Hnf4a CUT&Tag peaks were outside of ATAC-seq peaks in the same sample. I am assuming that you meant to say that the "The vast majority of both FoxA1 and Hnf4a binding sites in the induced sample matched inaccessible regions in the uninduced sample…"

Yes. We have clarified this in the revised text (line 141).

Please provide quality summary statistics for ATAC-seq and CUT&Tag data. Were these samples sequenced to a similar degree? Do the ATAC-seq samples have a similar proportion of reads surrounding TSSs, or within peaks? Were a similar number of peaks called in each ATAC-seq sample?

We have added Supplementary Files 4-5 to address these statistics and included the analysis within our methods (lines 410 and 436).

Pg 12, line 235. "cooperative" is a better word than "collaborative" to describe this phenomenon. Similarly, the term "collaboratively co-bound" is better represented using the term "cooperatively co-bound". Note that the term cooperative does not necessitate direct protein-protein interactions between TFs, but can rather merely reflect their cooperative ability to stabilize a nucleosome free stretch of DNA (PMID 21149679).

Thank you for the recommendation. We have adjusted our language throughout (lines 233 and 242, and after).

Please provide further description on how you define the list of organ-specific genes?

We have detailed how we defined tissue-specific genes in the Methods section (line 390).

Per convention, please use uppercase protein names as K562 is a human cell line.

We have corrected this error throughout.